# Exploring gestational age, and birth weight assessment in Thatta district, Sindh, Pakistan: Healthcare providers' knowledge, practices, perceived barriers, and the potential of a mobile app for identifying preterm and low birth weight

Shiyam Sunder Tikmani[1,2]*, Thomas Mårtensson[1], Sana Roujani[2‡], Anam Shahil Feroz[2,3‡], Ayshe Seyfulayeva[4‡], Andreas Mårtensson[1‡], Nick Brown[1], Sarah Saleem[2]

1 Department of Women's & Children's Health, Global Health & Migration Unit, Uppsala University, Uppsala, Sweden, 2 Department of Community Health Sciences, Aga Khan University, Karachi, Pakistan, 3 Institute of Health Policy, Management, and Evaluation, University of Toronto, Toronto, Canada, 4 National School of Public Health, The NOVA University of Lisbon, Lisbon, Portugal

☯ These authors contributed equally to this work.
‡ SR, ASF, AS and AM also contributed equally to this work.
* shiyam.sunder@kbh.uu.se

## Abstract

### Introduction

Reliable methods for identifying prematurity and low birth weight (LBW) are crucial to ending preventable deaths in newborns. This study explored healthcare providers' (HCPs) knowledge, practice, perceived barriers in assessing gestational age and birth weight and their referral methods for preterm and LBW infants. The study additionally assessed the potential of using a mobile app for the identification and referral decision of preterm and LBW.

### Methods

This qualitative descriptive study was conducted in Thatta District, Sindh, Pakistan. Participants, including doctors, nurses, lady health visitors, and midwives, were purposefully selected from a district headquarter hospital, and private providers in the catchment area of Global Network's Maternal and Newborn Health Registry (MNHR). Interviews were conducted using an interview guide after obtaining written informed consent. Audio recordings of the interviews were transcribed and analyzed using NVIVO® software with an inductive approach.

### Results

The HCPs had extensive knowledge about antenatal and postnatal methods for assessing gestational age. They expressed a preference for antenatal ultrasound due to the perceived

**Data Availability Statement:** All relevant data are within the manuscript.

**Funding:** The authors received no specific funding for this work.

**Competing interests:** The authors have declared that no competing interests exist.

accuracy, though accept practical barriers including workload, machine malfunctions, and cost. Postnatal assessment using the Ballard score was only undertaken sparingly due to insufficient training and subjectivity. All HCPs preferred electronic weighing scales for birth weight Barriers encountered included weighing scale calibration and battery issues. There was variation in the definition of prematurity and LBW, leading to delays in referral. Limited resources, inadequate education, and negative parent past experiences were barriers to referral. Foot length measurements were not currently being used. While mobile apps are felt to have potential, unreliable electricity supply and internet connectivity are barriers.

## Conclusion

The HCPs in this study were knowledgeable in terms of potential tools, but acknowledged the logistical and parental barriers to implementation

## Introduction

Sustainable Development Goal (SDG) 3.2.2 specifically targets reducing neonatal mortality and SDG 3.8 targets achieving Universal Health coverage. SDGs provide realistic aspirational targets for improving maternal and newborn outcomes at the national level by ensuring access to essential healthcare services [1–3]. Maternal and newborn health encompasses antenatal care, gestational age (GA) assessment, place of delivery, peri- and post-partum care, and the tailored management of Low Birth Weight (birth weight < 2,500 g) and prematurity (gestational age < 37 weeks).

Assessing gestational age and measuring the weight of the baby at birth is essential for determining the level of care needed [4]. However, in rural areas, HCPs face challenges in assessing GA and birth weight potentially leading to delays in identifying preterm and LBW babies, treatment, and, as a result, suboptimal time to referral for appropriate care.

Pakistan is committed to achieving its maternal and child health-related SDGs [5]. However, in rural Pakistan, low coverage of antenatal and postnatal care, insufficient maternal and newborn referral systems, and community health-seeking behavior are some of the main causes of late identification of preterm and LBW babies [6, 7].

To address the limited access to healthcare services in remote areas, mobile phone health-based applications have gained popularity [8]. These applications can be used for calculating drug doses, monitoring fetal growth during pregnancy, and more. Developing a mobile phone application that identifies GA and birth weight by measuring the foot length of newborns may have the potential for early identification and timely treatment of preterm and LBW babies.

The present study, performed in rural areas of the District Thatta in Sindh province of Pakistan aimed to: A) Explore HCP knowledge, practices and barriers in assessing pre- and postnatal GA, birth weight gestate and referral patterns when identified. and B) the attitude of HCPs regarding the potential use of a mobile phone application for identification of preterm and LBW babies.

## Materials and methods

### Study design

The study used a descriptive qualitative research design and conducted in-depth interviews (IDI) with HCPs to understand their knowledge, practices and perceived barriers in assessing

GA and birth weight, as well as their referral practices for preterm and LBW infants. It also explored the potential of using a mobile app for identifying and referring preterm and LBW cases.

## Study site

The study was undertaken in Thatta, a rural district, in Sindh Province of Pakistan from 1$^{st}$ March 2022 to 30$^{th}$ June 2022. According to the 2017 Pakistan census the total population of Thatta district was 9,821,383, with a population density is 114.6 people per square kilometer of which approximately 82.0% rural dwellers. Of these, 77.4% of adult females and 54.3% of males have not completed primary education [9]. Farming, fishing, and labor are the most common occupations of the population living in rural Thatta.

Study participants were recruited from the catchment area of the Global Network for Maternal Newborn Health Registry (MNHR) which provides population-based estimates of stillbirth, neonatal mortality, maternal mortality, prematurity, and LBW. In the MNHR, registry administrators enroll pregnant women in early pregnancy who are then followed up until the time of delivery. Subsequently, infant follow-up is undertaken until approximately 42 days post-partum. MNHR registry administrators calibrate weighing scales in both public and private sector hospitals once a week by a registry administrator [10]. In 2021 the rate of preterm births was 21.8% [11] and LBW was 21.4% in Thatta respectively [11]. Based on a previous study, 97% of women had at least one ANC visit, however, only 40% of women had 4 antenatal care (ANC) visits during their pregnancy in Thatta [12].

## Eligibility criteria

To enhance diversity, the sampling frame included doctors, nurses, lady health visitors, and community midwives (CMW). Detailed eligibility criteria are summarized in Table 1.

## Sampling technique and study participants

A purposive criterion sampling technique was used to recruit HCPs for in-depth interviews (IDI). Purposive sampling offers a multitude of sampling techniques that can be employed in qualitative research designs, presenting a significant advantage. These techniques encompass various approaches, including homogeneous sampling, critical case sampling, expert sampling, and others, thus providing a broad spectrum of options [13–15].

**Table 1. Eligibility criteria of Health Care Providers (HCPs).**

**Obstetrician**
• Postgraduate qualification (Diploma or fellowship in obstetrics and Gynaecology).
• At least one year of post-diploma/fellowship experience working in a rural setting.
**Pediatrician/neonatologist**
• Postgraduate qualification (Diploma or fellowship in pediatrics).
• At least one year post-diploma/fellowship experience working in a rural setting.
**Doctor**
• A medical doctor with six months internship in obstetrics and gynecology or pediatrics having at least one year of experience working in the rural setting.
**Lady Health Visitors or Community Midwives**
• Diploma in Midwifery or Lady Health Visitor.
• At least one year or more in the rural setting.

## Data collection procedure

The MNHR maintains a comprehensive list of HCPs, in public and private settings within its catchment areas. To verify the list of HCPs of public sector hospitals, MNHR registry administrators (RAs) contacted the hospital in charge. RAs obtained the recent list of HCPs working in their healthcare facilities, which included information such as age, gender, qualification, placement in a health facility (e.g., labor room, neonatal nursery), years of experience, and duration of employment in a current health facility. RAs then screened the HCPs for eligibility. Those found eligible were approached by RAs and asked if they were willing to participate in the study. For those who expressed willingness, the RAs scheduled the interview.

The interviews with HCP were conducted in Sindhi, in separate rooms in public and private healthcare facilities and clinics of CMW and lady health visitors. The researchers ensured the privacy and comfort of the data collection sites. IDIs were conducted by a male (Principal investigator-SST) researcher. A female notetaker was present to take interview notes. Written informed consent was obtained from HCPs to audio-record the interview using an audio recorder of a mobile phone and taking notes.

The demographic characteristics of study participants such as age, sex, designation, education, years of experience in total, and the current health facility were recorded. After exchanging initial pleasantries, the interviews were formally conducted using the interview guide. On average, the IDIs lasted for 40 minutes (ranging from 30 to 45 minutes).

## Interview guide

An interview guide for IDIs with probes was developed based on a literature search [5, 16–18] and experts' opinions in the field of qualitative research and reproductive health. The interview guide consisted of different sections; the importance of assessing GA and birth weight, methods, and barriers of assessing GA and birth weight, referral practices related to preterm and LBW, and the potential use of mobile phone applications for identification of preterm and LBW. A pretest of the interview guide was performed to identify any deficiencies that were corrected before the actual data collection (S1 File).

## Data analysis

Demographic information was entered into MS Excel ® spreadsheet for descriptive analysis. Qualitative data were analyzed using NVivo12 Plus® software using an inductive approach [19]. A trained transcriber translated and transcribed the audio recordings from Sindhi to the English language. The anonymity of the participants was ensured during the data analysis process by attributing a numerical identification code. Two investigators (SST & SS) independently read the transcripts several times to develop an interpretation of the participant's discussions. Data saturation was considered to be achieved when the investigators started to observe redundancy in the data, meaning that new interviews yield no additional information or insights. In this study, data saturation was achieved on the 15[th] interview. The text was divided into 'meaningful units' which were shortened and labeled with a 'code' with mutual agreement between the two investigators while maintaining the study context. The codes were then analyzed and organized into categories to capture the manifest meaning. Themes and subthemes were identified as a final step [20].

## Ethical consideration

This study was approved by the Ethical Review Committee of Aga Khan University, Karachi (Reference number: 2022-7068-20733). Written informed consent was obtained from all the participants of the study.

**Table 2. Characteristics of study participants (n-15).**

| Median age (Range) | 38 (Range 22–59) years |
|---|---|
| Gender | |
| •Female | 13 (86.7%) |
| •Male | 2 (13.3%) |
| Medical cadre | |
| Doctors | 6 (40.0%) |
| •Community mid-wives | 5 (33.3%) |
| •Nurses/lady health visitors | 4 (26.7%) |
| Years of experience | 12.1 (1–26 years) |
| Median (Range) | |

## Results

A total of 15 IDIs were conducted of which 6 involved doctors, 4 nurses/ lady health visitors and 5 midwives. The participant's profile is given in Table 2.

### Gestational age and birth weight assessment

**Importance of gestational age and birth weight.** HCPs recognized the significance of assessing GA and birth weight due to their impact on child management and survival. They recognized the importance of GA assessment for insights into progress, determination of the estimated delivery date and the facilitation of preparing families for transfer. Also, birth weight in the labor room enables us to refer the baby to a neonatal nursery.

A doctor shared her thoughts on gestational:

"*By this, we get her Expected Date of Delivery and the duration of pregnancy, the family can be then mentally prepared for arranging their transport and to plan savings.*" *(Doctor)*

Doctors shared her thoughts on birth weight:

"*Taking birth weight is important. If we don't the birth weight and the baby is sick, then the baby may not get the correct drug dosage.*" *(Doctor)*

**Preferred method of assessment of gestational age by HCP.** HCPs employed different methods to determine antenatal GA. Doctors prefer antenatal ultrasound, followed by LMP, and clinical examination, while CMW and lady health visitors prefer LMP, followed by antenatal ultrasound, and fundal height. Some providers measure GA in months rather than weeks, mainly relying on LMP. One medical doctor mentioned.

"*I prefer ultrasound for estimating gestational age as it also provides additional information such as movement of baby, position and any complications in the baby.*" *(Doctor)*

### Prenatal methods of gestational age assessment

**Knowledge and practices related to ultrasound.** HCPs generally advised utilizing antenatal ultrasounds for accurate dating, with first-trimester ultrasounds deemed the most reliable method for estimating the delivery date. However, some providers preferred ultrasounds for assessing GA, heart movement, baby position, and identifying anomalies after 20 weeks.

*"In the first trimester, we do an ultrasound to check the baby's growth and position. Ultrasounds are done at 12, 24, and 28 weeks unless there are complications. Sometimes parents request an ultrasound to learn about the sex of the baby only". (Obstetrician)*

**Knowledge and practice related to the last menstrual period.** HCPs such as CMW, and lady health visitors commonly rely on LMP to estimate GA and calculate the estimated date of delivery. However, they know that LMP is not reliable. They also acknowledged that many women cannot recall it accurately and often relate their LMP with cultural or seasonal events, moon cycle, and harvest season.

A doctor shared:

*"Pregnant women most of the time relate their LMP with harvesting season of wheat, rice as well as major local events such as ´Urs´, the celebration of a famous saint" (Doctor)*

**Knowledge and practices related to fundal height.** Fundal height is measured during ANC check-ups to estimate GA and monitor the baby's growth. It is used when LMP is unknown, or ultrasound is unavailable. HCPs' level of confidence in this method varies, they mostly count the number of fingers from the umbilicus to fundal height to determine fundal height and visa-à-vis GA.

A doctor shared that:

*"Fundal height is better for both gestational age and growth of the baby. But, if baby's growth is compromised, fundal height could be less reliable." (Doctor)*

## Barriers to prenatal methods of assessing gestational age

**Barriers to using ultrasound examination.** HCP cited barriers to ultrasound usage: workload, power outages, machine malfunctions, limited sonographers, supply shortages, high private facility costs, lack of training and maintenance, and quality control concerns. Some of the quotes by the respondents given below express the difficulties of using ultrasound for gestational dating.

*"Although we have generator support, power failures during ultrasound examinations can lead to loss of concentration and hinder the quality of the exam." (Nurse)*

**Barriers to using LMP.** According to HCP, the barriers to reporting precise LMP included recall, cultural norms, absence of ANC records, language barriers, and factors such as being uneducated, tribal, or being a woman who is shy about discussing menstruation or remaining sexually active in older age.

*"In our practice, newlywed girls or women with frequent pregnancies often don't disclose their last menstrual period due to shyness. Additionally, some pregnant women have irregular menstrual cycles due to hormonal imbalances, therefore unable to recall their LMP." (Doctor, and Community midwife)*

**Barriers to using the fundal height method.** The HCP verbalized that the main barriers are inadequately trained staff, women feeling discomfort during the examination, shyness during abdominal exposure, and women with medical conditions. An obstetrician shared that:

*"A woman came without an ultrasound examination and was unable to recall LMP despite every effort. I tried to take fundal height but was unable to locate the fundus due to obesity."* *(Doctor-Obstetrician)*

## Postnatal methods of assessing gestational age

**Knowledge and practices Ballard scoring.** In the nursery unit of a secondary care hospital, we observed post-natal Ballard examination charts and feeding protocols for preterm births displayed on walls. However, staff rarely utilized these job aids, except for doctors who used the preterm feeding protocol. When asked about confirming or assessing the baby's GA upon referral, the doctor responded:

*". . .only ultrasound of booked cases is available, and we don't have formal training to do postnatal Ballard scoring. Therefore, it is very difficult for us to estimate gestational age" (Doctor)*

**Knowledge and practices of foot length.** Only a few of the HCPs know the use of a newborn's foot length to calculate the GA. However, none of the HCPs knew that foot length may indicate the birth weight of the baby. After an explanation, HCP found foot length measure as a simple and objective method of assessing GA.

*"I heard that foot length and other body part measurements are used to assess gestational age, but I don't practice it because I rely on ultrasound results which are accurate and reliable."* *(Doctor)*

## Assessing birth weight

**Preferred method of assessment of birth weight by HCP.** HCPs preferred electronic weighing scales for taking birth weight. None of the HCPs in this study utilized any other method of taking birth weight. However, some HCPs knew of other methods such as manual weighing scales, and hanging weighing scales, though, felt them to be inaccurate.

*"We are using electronic weighing scale because it's easy and accurately measure birth weight"* *(Doctor, CMW 2, 4)*

## Practices for taking weight

All HCPs verbalized that before weighing the baby, they make sure that the digital weighing scale shows zero reading. HCPs working in the labor room verbalized that the baby weighed without clothes. Some HCPs shared that they weigh the baby with a cloth nappy. In case the machine shows errors, the staff changes the batteries or sends scales for repair or replacement.

*"In the labor room, we make sure that the weighing scale does not show error. Once the machine shows zero, we measure the baby without clothes. However, small sick babies were wrapped in cotton therefore birth weight may be slightly very."* *(Doctor)*

**Barriers to taking birth weight.** HCPs verbalized that lack of a standard electronic weighing scale, lack of calibration, and batteries run out are common barriers to taking the birth weight.

"*..in many instances when the batteries run out leading to inaccurate weight baby is looking small but the machine is showing normal birth weight.*" (Nurse)

## Knowledge and practices for preterm & low birth weight babies

HCPs have different understandings of the definitions of preterm and LBW, with only doctors and nursing staff having a clear understanding of the definitions. Some HCPs used the terms preterm and LBW interchangeably, while some believe they can identify preterm babies based on clinical examination. According to doctors:

"*When a baby is born before 37 weeks is called preterm. Since they are born preterm, so they are low birth weight too.*" (Doctors 1 & 3)

While according to a CMW:

"*The preterm baby is born before 34 weeks or weighs < 1.5 kg at birth. Normal birth weight is 2.5 kg. Some preterm babies may have reddish shiny skin covered with thin hair, slow respiration, and low body temperature.*" (CMW, 2, 3, 5)

**Practices related to the management of preterm and LBW in the labor room.** HCPs were aware of the care provided to preterm babies soon after birth. Most of the HCPs in the labor room take the APGAR score at one minute, apply a cord clamp, give vitamin K, and refer the baby to the nursery for the opinion of a doctor in the nursery. Sometimes doctors refer the term good weight babies to the nursery for care.

"*First aid is provided to the baby, including cleaning, drying, warming, ensuring a clear airway, oxygen if necessary, and referral to the nursery for child specialist's opinion.*" (LHV, CMW, Doctor)

**Practices related to the management of preterm and LBW in the nursery.** HCPs in the nursery, examine and admit the baby, if needed, check the APGAR score at five minutes, monitor glucose levels, and start administering antibiotics. Moreover, HCPs provide phototherapy, incubator care, and intravenous fluids, refer the baby, and counsel the parents if needed.

"*I face difficulty in assessing GA and making treatment decisions when the mother's ultrasound is not available. Then we rely on the history provided by attendants and treat the baby accordingly.*" (LHV)

**Practices regarding referral of preterm and LBW.** Nursery doctors or staff counsel parents, make referral letters and arrange ambulances despite the lack of a referral protocol for sick babies. CMW and Lady Health Visitors face challenges in the referral process but refer babies to the secondary hospital. For advanced care, a doctor refers the baby to another facility. CMWs and LHVs arrange ambulance transport without providing referral letters or contacting the referral hospital's healthcare provider.

"*We refer the sick preterm or LBW babies to ICU care with a proper referral letter and arrange free ambulance service. The ambulance is well-equipped with oxygen connection. The ambulance staff are also trained to manage sick babies en route.*" (Doctor nursery)

**Barriers to referral.**    HCPs note several referral barriers: parents' education, lack of resources/ambulance access in remote areas, family influence, lack of understanding, and previous bad experiences.

"*Parents agreed to move baby, but grandfather refused for financial reasons. The baby died during the discussion*" *(Doctor)*

## Potential usefulness of mobile-based applications

**Importance of mobile-based applications in healthcare.**    All HCPs felt that mobile phone applications for GA are potentially useful tools, apart from social media and WhatsApp, to explore recent literature and treatment guidelines for specific conditions. Moreover, HCP expressed that mobile phone applications could be a valuable tool for accurately calculating drug dosage to enhance patient safety and reduce the risk of adverse events. One of the obstetricians shared:

"*I am using mobile-based applications to read guidelines such as treatment of anemia and calculating the dose of drugs such as IV iron because it may cay cause any reaction*". She also added *"Most of the educated women are using mobile applications for tracking tetanus vaccination and anomaly scan."* *(Doctor)*

**Importance of potential mobile-based application for calculating gestational age and birth weight.**    HCP expressed that mobile applications could be valuable for calculating GA and birth weight, benefiting clinical management. They highlighted its potential usefulness for staff working in the evening, at night, or in remote areas where ultrasound and other assessment methods are unavailable. One doctor said:

"*This mobile application would be particularly useful and beneficial for Basic Health Unit-level facilities and small healthcare facilities where ultrasound facilities may not be available it would impact the treatment plan of preterm and LBW babies such as referral."* *(Doctor)*

**Using foot length as input to mobile application for gestational age calculation.**    HCP recognized the value of foot length as a simple and objective method for assessing preterm and LBW. They emphasized the importance of training in foot length measurement and using mobile applications when ultrasound is unavailable, especially during evening/night shifts. The mobile app is particularly useful when ultrasound reports are absent and for accurately determining GA in healthy preterm babies. One doctor said:

" *The application could prove useful when women present in labor with late, unreliable ultrasound reports or when women with diabetes give birth to babies that appear healthy but are preterm."* *(Doctor)*

**Barriers to the use of the mobile application.**    All HCPs mentioned that power failure, internet connectivity, expensive mobile internet packages with slow speed, and apps used only with the internet are the main barriers. Some HCPs also shared that if HCPs are not adequately trained the use of this application is not fruitful.

"*. . . mobile phones are not charged, and the internet is very slow. However, the offline application could be useful, especially in remote areas without internet facility*" *(CMW, Doctor)*

## Discussion

This study found that, for GA, doctors relied on antenatal ultrasound for its accuracy and availability CMWs and lady health visitors used LMP due to the absence of ultrasound facilities, referring women to private sonographers at their own expense. Power failures and cost were identified as barriers to ultrasound use. LMP reliability was questioned due to recall issues and women's educational status. In a study from Rajasthan, India [16], HCPs used a combination of LMP, fundal height, and ultrasound for GA assessment. Some relied on LMP or fundal height, despite their unreliability [17], while others used fundal height to confirm LMP. Antenatal ultrasounds were not offered directly, and referrals were made to higher-level facilities in the second or third trimester [18]. Also, providers deemed antenatal ultrasounds unnecessary for normal healthy pregnancies, mentioning factors such as long travel distances, private provider costs, and extended waiting periods in public hospitals. Fundal height measurement was least preferred due to inadequate training and maternal obesity. In another study, fundal height was the most frequently used method for GA assessment, providing estimates in months [21].

HCPs found Ballard scoring complex and burdensome, relying on GA from the mother's files. Foot length as a GA assessment tool was recognized but not practiced due to a lack of knowledge. Doctors facilitated referrals, but CMW caused delays due to limited access and lack of awareness. Referral letters were not always provided, but ambulances were arranged [22]. All HCPs used electronic weighing scales due to their accuracy. However, calibration and battery issues were the main barriers. In a study, it was reported that poor device condition, imprecise measurement practices, and battery issues are common barriers to accurate birth weight. We additionally found that babies are being weighed with no or minimal clothing for accurate weight. Since in the labor room, the environment for the examination of babies is warm it's feasible in labor rooms and neonatal nurseries. To prevent hypothermia small sick infants were wrapped in cotton and birth weight may not be accurate.

HCPs acknowledged the potential of mobile phone applications for GA calculation and identifying preterm and LBW infants. Barriers such as power failures and internet connectivity hindered their widespread use. Offline mobile applications were favored in remote areas. Ongoing research in Thatta district, Pakistan, aims to evaluate the use of foot length as a simple and accurate tool for assessing preterm and LBW infants.

The role of HCP is crucial in preventing neonatal mortality. Therefore, they must possess comprehensive knowledge of ANC practices, including GA assessment, regular check-ups for fetal growth, screening for potential health concerns, and education on healthy pregnancy practices [4]. Moreover, understanding the definitions of preterm birth and LBW, along with their associated risk factors and complications, is vital for managing preterm birth and LBW babies [23].

The study has potential limitations that should be acknowledged. Firstly, the selection of HCPs from a limited number of facilities in Thatta district, Pakistan may limit generalizability to other regions with different healthcare systems and cultural contexts. Secondly, reliance on self-reported data introduces the possibility of recall and social desirability biases. Furthermore, patient perspectives were not considered, focusing solely on HCP. Lastly, the study's qualitative nature and lack of quantitative data restrict statistical associations and comprehensive measurements.

In conclusion, the study found varying preferences among different HCP cadres for assessing GA, with a consistent preference for electronic weighing scales to determine birth weight. Discrepancies in prematurity and LBW definitions led to referral delays. Recommendations include investing in healthcare infrastructure, ensuring well-equipped maternity homes and

trained professionals, and launching awareness campaigns for prenatal care and early detection of preterm and LBW using mobile applications. Implementing these actions will overcome barriers and improve the identification and better feral decisions of premature and LBW infants in rural Pakistan.

## Supporting information

**S1 Checklist. Human participants research checklist.**
(DOCX)

**S1 File. IDI guide and patient characteristics.**
(DOCX)

## Acknowledgments

We acknowledge all study participants for their valuable participation and the dedicated support of the staff during the study. I dedicate this manuscript to my late father Professor Pirbhulal Tikmani and my Late mother Asha Devi (Revti).

## Author Contributions

**Conceptualization:** Shiyam Sunder Tikmani, Thomas Mårtensson, Andreas Mårtensson, Nick Brown, Sarah Saleem.

**Formal analysis:** Shiyam Sunder Tikmani, Sana Roujani, Sarah Saleem.

**Investigation:** Anam Shahil Feroz.

**Methodology:** Shiyam Sunder Tikmani, Ayshe Seyfulayeva, Andreas Mårtensson, Nick Brown, Sarah Saleem.

**Project administration:** Shiyam Sunder Tikmani, Sana Roujani, Anam Shahil Feroz.

**Supervision:** Thomas Mårtensson, Sana Roujani, Anam Shahil Feroz, Andreas Mårtensson, Nick Brown, Sarah Saleem.

**Validation:** Thomas Mårtensson, Ayshe Seyfulayeva.

**Writing – original draft:** Shiyam Sunder Tikmani.

**Writing – review & editing:** Shiyam Sunder Tikmani, Thomas Mårtensson, Anam Shahil Feroz, Ayshe Seyfulayeva, Andreas Mårtensson, Nick Brown, Sarah Saleem.

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
