## [Decision Letter · Decision Letter 0]

28 Nov 2023

PONE-D-23-24083Exploring gestational age, and birth weight assessment in Thatta district, Sindh, Pakistan: healthcare providers' knowledge, practices, perceived barriers, and the potential of a mobile app for identifying preterm and low birth weightPLOS ONE

Dear Dr. Tikmani,

Thank you for submitting your manuscript to PLOS ONE. After careful consideration, we feel that it has merit but does not fully meet PLOS ONE’s publication criteria as it currently stands. Therefore, we invite you to submit a revised version of the manuscript that addresses the points raised during the review process.

We look forward to receiving your revised manuscript.

Kind regards,

Sidrah Nausheen, FCPS

Academic Editor

PLOS ONE

Reviewers' comments:

Reviewer's Responses to Questions

**Comments to the Author**

1. Is the manuscript technically sound, and do the data support the conclusions?

Reviewer #1: Yes

Reviewer #2: Yes

2. Has the statistical analysis been performed appropriately and rigorously? 

Reviewer #1: Yes

Reviewer #2: N/A

3. Have the authors made all data underlying the findings in their manuscript fully available?

Reviewer #1: Yes

Reviewer #2: Yes

4. Is the manuscript presented in an intelligible fashion and written in standard English?

Reviewer #1: Yes

Reviewer #2: Yes

5. Review Comments to the Author

Reviewer #1: Simple and well written article. Good sample size with in depth interviews

quotes are well defined considering qualitative study.

No grammatical error. Methodology looks fine.

I did not find any major issue in study conduct .

Reviewer #2: The study is well written. However, the authors should give the description of the type of mobile app that was used to explore its potential in the current study. e.g. is it a standard mobile app, its name, was this app created for this study, what information is required in it to get the desired output etc.

6. PLOS authors have the option to publish the peer review history of their article (what does this mean?). If published, this will include your full peer review and any attached files.

Reviewer #1: No

Reviewer #2: No

---

## [Author Response · Author response to Decision Letter 0]

16 Jan 2024

PONE-D-23-24083 (Separate file is attached as well)

“Exploring gestational age, and birth weight assessment in Thatta district, Sindh, Pakistan: healthcare providers' knowledge, practices, perceived barriers, and the potential of a mobile app for identifying preterm and low birth weight.”

The point-by-point responses to reviewers’ comments

On behalf of all authors, I express my sincere gratitude for your thorough review of our research paper, and we truly appreciate the time and expertise you dedicated to the process. Your contributions are invaluable, and we are grateful for your commitment to advancing academic excellence.

We've made revisions using the "Track Changes" function in the MS Word document, clearly marked as such. 

Additionally, we've included a clean copy following the acceptance of track changes.

Reviewer's comments Author's response(s)

Reviewer #1: Simple and well-written article. Good sample size within depth interview quotes are well defined considering qualitative study. No grammatical errors. The methodology looks fine. I did not find any major issues in the study conduct. 

Response: Thank you for the comments. 

Reviewer #2: The study is well written. However, the authors should describe the type of mobile app that was used to explore its potential in the current study. e.g. is it a standard mobile app, its name, was this app created for this study, what information is required in it to get the desired output, etc. Response: Thank you for your comment. In our study, we did not utilize a specific mobile app to explore its potential. We explored the opinions of health providers on a mobile phone application for the identification of preterm and low birth weight.

However, in the future, we plan to conduct a quantitative study to develop a regression predictive model. Based on this model we will develop an app. This app will use post-natal foot length as an input.

---

## [Editor Report · Decision Letter 1]

9 Feb 2024

Exploring gestational age, and birth weight assessment in Thatta district, Sindh, Pakistan: healthcare providers' knowledge, practices, perceived barriers, and the potential of a mobile app for identifying preterm and low birth weight

PONE-D-23-24083R1

Dear Dr. Tikmani,

We’re pleased to inform you that your manuscript has been judged scientifically suitable for publication and will be formally accepted for publication once it meets all outstanding technical requirements.

Kind regards,

Sidrah Nausheen, FCPS

Academic Editor

PLOS ONE
---

## [Editor Report · Acceptance letter]

29 Mar 2024

PONE-D-23-24083R1 

PLOS ONE

Dear Dr. Tikmani, 

I'm pleased to inform you that your manuscript has been deemed suitable for publication in PLOS ONE. Congratulations! Your manuscript is now being handed over to our production team.

Kind regards, 

on behalf of

Dr. Sidrah Nausheen 

Academic Editor

PLOS ONE